# Pharmacy Workload in Clinical Trial Management: A Preliminary Complexity Assessment Tool for Sponsored Oncology and Haematology Trials

**Lorenzo Gasperoni** [1,*] **, Carla Masini** [1] **, Giada Toscano** [1] **, Alessandro Cafaro** [1] **, Chiara Zani** [1] **, Cristina Andrianò** [1] **, Paolo Silimbani** [1] **, Caterina Donati** [1] **, Giorgia Bortolin** [2] **and Sara Cecco** [2]

1   Oncological Pharmacy Unit, IRCCS Istituto Romagnolo per lo Studio dei Tumori (IRST) "Dino Amadori", Emilia Romagna, 47014 Meldola, Italy; carla.masini@irst.emr.it (C.M.); giada.toscano@irst.emr.it (G.T.); alessandro.cafaro@irst.emr.it (A.C.); chiara.zani@irst.emr.it (C.Z.); cristina.andriano@irst.emr.it (C.A.); paolo.silimbani@irst.emr.it (P.S.); caterina.donati@irst.emr.it (C.D.)
2   Hospital Pharmacy Unit-CRO Aviano, National Cancer Institute IRCCS, Friuli-Venezia Giulia, 33081 Aviano, Italy; giorgia.bortolin@cro.it (G.B.); scecco@cro.it (S.C.)
*   Correspondence: lorenzo.gasperoni@irst.emr.it

**Abstract:** Investigational drug services need to be organised in a structured approach, especially for sites with a large number of ongoing clinical trials. The aim of this study was to develop a tool to assess the complexity of pharmacy involvement in a sponsored oncology clinical trial. Categorisation into ordinal complexity categories was used to assess the complexity of the clinical trials for consistent pharmacy grant applications. The 15 items of the tool were divided into three sections, and individual item scores were agreed upon among four pharmacists with experience in the conduct of clinical trials at two different centres. A final version of the tool, named Pharm-CAT, was approved. The pharmacists were instructed to use Pharm-CAT to assign a score to each new sponsored trial. To determine the cut-offs for the complexity categories, the scores were sorted in ascending order and the cut-offs corresponding to the first and third tertiles of the score distribution were selected. To verify the reproducibility of the results, Pharm-CAT was applied by two pharmacists independently for each trial. Pharm-CAT proved to be user-friendly. Sixty clinical trials were evaluated and a total of 120 scores were recorded. Low-complexity scores ranged from 0 to 19, medium-complexity scores ranged from 20 to 25, and high-complexity scores were 26 or higher. The average score recorded was 22.88 points. Prospective multicentre validation of Pharm-CAT is needed to confirm its applicability.

**Keywords:** clinical trial; pharmacy; workload; oncology; scoring method

## 1. Introduction

The development of new drugs in recent years, as well as new regulations and guidelines for research, has led to important changes in the way research is conducted and the role of each professional [1,2]. Cancer clinical trials are becoming increasingly complex, requiring the involvement of multiple disciplines and dedicated personnel to perform clinical, regulatory, and administrative activities and protocol patient-related procedures [3–5]. Clinical trials are designed to produce results in a short time, speeding up the approval of new molecules [1]. The total number and frequency of procedures, tests, data collections, and data elements specified in each clinical trial protocol affect the effort required by a trial site to ensure compliance with the protocol and regulatory requirements. According to good clinical practice (GCP), the principal investigator (PI) may/should delegate some or all of the investigator's responsibilities for investigational medicinal products (IMPs) to a trained pharmacist or other appropriate person [6]. This results in the lack of an internationally standardised professional profile, and the different roles that pharmacists play in the research team could make it difficult to predict the workload in another context.

In Italy, the delegation of this responsibility to a pharmacist is allowed, but not mandatory. The Italian Medicines Agency requires the presence of a pharmacist responsible for the management of IMPs only for phase I trials [7]. There are differences in the organisation and staffing of each pharmacy service. In some units, staff members working in the clinical trial area are involved in other hospital pharmacy activities during the working day, whereas in other centres, their involvement is exclusive [8]. Most pharmacies provide basic services for research, such as dispensing and stock control, while more specialised activities are common at sites with a greater commitment to research [1].

The responsibilities of the investigational drug services include, but are not limited to, the study feasibility, a site initiation visit, the training of the involved personnel, a receipt for the IMPs, accountability, storage, the calibration and maintenance of temperature-monitoring equipment, dose preparation (under sterile conditions and ensuring blinding, if required), dispensing, the recording of drugs returned by patients, the final disposition of drugs, the return to the sponsor, a close-out visit, and the hosting of monitors and auditors. All of these activities must be carried out while ensuring safety, compliance with legal and regulatory requirements, adherence to internal standard operating procedures and policies, compliance with the protocol, and the quality of the trial process. Most drug management tasks are considered "source documents" that must be available for monitoring visits, sponsor audits, and regulatory inspections [6], so it is crucial that they are performed and documented to ensure compliance. For these reasons, the role of the pharmacy is important in adding quality and value to clinical trials. It should also be noted that the management of IMPs in oncology clinical trials is becoming increasingly sophisticated, with complex preparations (high-risk compounding, blinding), complex dosing regimens, multiple drug regimens, a high toxicity potential, extensive sponsor training modules, and increasingly stringent sponsor requirements for storage and monitoring. In this context, investigational drug services are time- and effort-consuming, vary from trial to trial depending on their complexity, and need to be organised in a structured approach, especially for sites with a large number of ongoing clinical trials. In light of these considerations, the involvement of pharmacy staff needs to be measured and evaluated.

Pharmacy costs are not routinely budgeted, nor do they need to be budgeted. This results in situations where the research pharmacy costs are absorbed by the institution or are covered by other means, such as a core grant [1]. Alternatively, when pharmacy costs are budgeted, the most common system used is a fixed percentage for all clinical trials [8,9]. A system based on the payment of pharmacy grants according to the complexity of the clinical trial from the perspective of the pharmacy service should be considered. The system should assess the complexity of a trial based on the involvement of the pharmacy in order to provide a consistent grant at the contractual stage between the sponsor and the research site. The complexity of a clinical trial should be assessed at an early stage, before the contract is signed, using a scoring tool. The numerical value of complexity scores can be difficult to interpret and apply in a meaningful way. Categorising the tool scores into ordinal complexity categories would be more intuitive and serve as a practical and effective tool for assessing the complexity of clinical trials and applying for consistent pharmacy grants [2]. Grants should be recognised when the site is activated, regardless of whether patients are enrolled in the trial, and for each patient enrolled. In this way, revenue generated by clinical trials should be proportionate to the impact on the pharmacy service's workload, which would result in a more accurate quantification than applying a simple percentage of the amount received by the principal investigator for each trial patient.

There is limited research available that has measured the complexity of pharmacy services for clinical trials. Some of these works are based on the use of tools that measure the time spent on specific tasks or the resources of the pharmacists or professionals employed [1,8,9]. The number of protocols a pharmacist can manage depends on many factors: the complexity of the protocols, the responsibilities of the pharmacy staff, the experience of the staff, and the organisation of the research centre [1]. As a result, it is difficult to translate the complexity of clinical trials from the perspective of the pharmacy service

into resources in terms of staff and time. In the work by Pagès-Puigdemont et al., routine activities were measured in time and then translated into value, but non-routine activities were voluntarily excluded. This resulted in a high value being assigned to activities that are always performed and time-consuming, but can be planned and, therefore, have less impact on a pharmacist's work, such as site selection visits and site initiation visits. Values were not assigned to non-routine activities that, despite having less impact in terms of time, are outside the scope of routine activities and are more critical for the patient and/or for the proper conduct of the trial [8]. Song et al. developed a systematic complexity scoring tool to assess pharmacy effort, but the development of the tool required application during the study period, limiting the ability of the tool to categorise the study in the preliminary phase and influence the contract [10]. The literature suggests that the largest number of high-complexity clinical trials are in oncology [8,9] due to the specific trial designs (basket and umbrella trials are clear examples), patients, and drug management. In addition, academic trials are often pragmatic and closely resemble daily clinical practice, with broader eligibility criteria, less frequent disease assessments, fewer data requests, less frequent monitoring visits, fewer bureaucratic constraints, fewer training documents, and a smaller training workload [5,11]. Therefore, for an oncology research centre running multiple clinical trials simultaneously, it is necessary to build a tool with complexity categories that are calibrated to oncology-sponsored trials, thereby increasing the sensitivity of the tool. The aim of this study was to develop a tool to assess the complexity of pharmacy involvement in a sponsored oncology or haematology clinical trial. The tool score cut-off points for the complexity categories (low, medium, and high) were then identified. Categorisation into ordinal complexity categories will serve as a practical and effective tool for assessing the complexity of clinical trials for consistent pharmacy grant applications.

## 2. Materials and Methods

The items of the tool and the individual item scores were agreed upon among the pharmacists at two different cancer research centres. It was decided that at least two pharmacists from each centre would be involved in the construction of the tool to ensure that all potential items of interest, regular activities, and extraordinary activities were brought to light. Disagreements regarding the definition of different items were discussed and a final version of the pharmacy complexity assessment tool (Pharm-CAT) was approved.

Each of the selected items had a concrete impact on the workload of the pharmacy service or had an impact by deviating from standard procedures. For example, if a drug is not provided by the sponsor and has to be provided by the trial site, it is advisable to separate a single batch for use in the trial, label it in accordance with GCP, and store it in the clinical trial area to ensure the traceability of the drug. Activities that had an impact on the workload, but were common to all the sponsored clinical trials were deliberately excluded, e.g., feasibility assessments, a site initiation visit, monitoring visits, and training activities.

Pharm-CAT consisted of 15 items divided into three sections: study design, drug management, and drug preparation. The score assigned to each item ranged from 0 to 3 points. Items 1, 3, 4, 5, 7, 12, and 13 received 1, 2, or 3 points; items 2 and 6 received 2 or 3 points; item 8 received 1 or 2 points; items 9, 10, and 11 received 1 or 3 points; item 14 received 0, 2, or 3 points; and item 15 received 0 or 3 points. In order to obtain a numerical complexity score, the total score for each clinical trial was calculated by adding the scores of the different items according to the scale shown in Table 1. The resulting score was a minimum of 15 points and a maximum of 44 points. The pharmacists were instructed to use Pharm-CAT to assign a score to each new sponsored trial. Each assessment generated by the compilation of Pharm-CAT for a new sponsored clinical trial from July 2023 to February 2024 was archived after being recorded in an Excel data collection table containing the following data: the research centre, trial acronym, trial type (oncology or haematology and phase I, II, or III), assessing pharmacist, and score. The Pharm-CAT scores were divided

into 3 categories: low-complexity, medium-complexity, and high-complexity. To determine the cut-offs for the three complexity categories, we sorted the scores in ascending order and selected the cut-offs corresponding to the first and third tertiles, i.e., at 33.3% and 66.7% of the score distribution. Ensuring reproducibility is an important step in increasing the reliability and usefulness of Pharm-CAT. To verify reproducibility, Pharm-CAT was applied by two pharmacists independently for each new clinical trial to highlight any differences in the total score and in the assigned complexity category. Independent scoring is essential for establishing the accuracy of Pharm-CAT. To establish the complexity categories and check the reproducibility, we determined that at least 50 trials should be independently scored by two pharmacists to record 100 scores. Once the cut-offs were established, Cohen's linear weighted kappa was used in the cross-classification as a measure of the agreement among pharmacists. The characteristics of the data contributed during the project period were summarised using percentages.

**Table 1.** Pharmacy complexity assessment tool (Pharm-CAT).

| | Items | Score |
|---|---|---|
| **Study design** | **1. Study phase**<br>1 point—phase II or phase III<br>3 points—phase I | |
| | **2. Type of drug**<br>2 points—oral drug<br>3 points—injectable drug | |
| | **3. Interactive web response systems (IWRSs)**<br>1 point—consultation only (documentation and reporting)<br>2 points—supply chain management<br>3 points—treatment assignment | |
| | **4. Number of drugs involved**<br>1 point—1 drug<br>2 points—2 drugs<br>3 points—$\geq$ 3 drugs | |
| | **5. Days of therapy per cycle**<br>1 point—1 day<br>2 points—2 or 3 days<br>3 points—$\geq$ 4 days | |
| **Drug management** | **6. Pharmacy staff**<br>2 points—logistics operator + pharmacist<br>3 points—logistics operator + pharmacist + technical operator | |
| | **7. Storage conditions**<br>1 point—controlled room temperature or under refrigeration<br>2 points—controlled room temperature and under refrigeration<br>3 points—deep-freeze | |
| | **8. Supply of special material/medical devices**<br>1 point—no<br>2 points—yes | |
| | **9. Relabelling of drugs not supplied**<br>1 point—no<br>3 points—yes | |
| | **10. Returned drug accountability**<br>1 point—no<br>3 points—yes | |

**Table 1.** *Cont.*

| Items | Score |
|---|---|
| **11. Drug resupply**<br>1 point—automatic<br>3 points—manual | |
| **12. Blindness**<br>1 point—no<br>2 points—double-blind<br>3 points—pharm-unblinded | |
| **13. Dose preparation**<br>1 point—ready-to-use drug<br>2 points—personalized dose<br>3 points—reconstitution of the drug + personalized dose | |
| **14. Drug preparation procedure**<br>2 points—prohibition of use of closed system transfer devices<br>3 points—preparation time exceeding 20 min<br>0 points—all other cases | |
| **15. Personalized administration**<br>3 points—personalized instruction (dose-preparation-based)<br>0 points—all other cases | |
| **Total score** | |

*(Left side row grouping label: **Drug preparation**)*

Each pharmacist involved in the construction and use of the tool was a professional with proven experience in the management of experimental drugs in oncology and/or haematology clinical trials.

### 3. Results

Pharm-CAT took 3–5 min to complete. Sixty clinical trials were evaluated and a total of 120 scores were recorded. Centre 1 evaluated 40 trials (80 scores) and centre 2 evaluated 20 trials (40 scores). In 77% of the studies, the independent assessment by the two pharmacists resulted in the same score; in 14 studies, the score was different. The difference was only one point in 11 studies, two points in 2 studies, and four points in 1 study. The average score difference in 60 trials was 0.32 points.

The result of the 120 scores determined the cut-offs, resulting in the three complexity categories. Low-complexity scores ranged from 0 to 19, medium-complexity scores ranged from 20 to 25, and high-complexity scores were 26 or higher. The average score recorded was 22.88 points. The lowest score recorded was 15 points and the highest was 33 points. The Cohen's weighted kappa calculation was 0.98, confirming agreement among pharmacists. Table 2 shows the categories of the evaluated clinical trials.

**Table 2.** Results of the evaluated clinical trials.

| | Clinical Trials | % | Average Score | Complexity | | | | | |
|---|---|---|---|---|---|---|---|---|---|
| | | | | Low | % | Medium | % | High | % |
| **Total** | 60 | 100.00 | 22.88 | 18 | 30.00 | 23 | 38.33 | 19 | 31.67 |
| Centre 1 | 40 | 66.67 | 22.50 | 14 | 35.00 | 13 | 32.50 | 13 | 32.50 |
| Centre 2 | 20 | 33.33 | 23.63 | 4 | 20.00 | 10 | 50.00 | 6 | 30.00 |
| Haematology | 21 | 35.00 | 23.98 | 6 | 28.57 | 5 | 23.81 | 10 | 47.62 |
| Oncology | 39 | 65.00 | 22.28 | 12 | 30.77 | 18 | 46.15 | 9 | 23.08 |
| Phase I | 6 | 10.00 | 24.92 | 0 | 0.00 | 3 | 50.00 | 3 | 50.00 |
| Phase II | 18 | 30.00 | 21.78 | 10 | 55.56 | 2 | 11.11 | 6 | 33.33 |
| Phase III | 36 | 60.00 | 23.08 | 8 | 22.22 | 18 | 50.00 | 10 | 27.78 |

Thirty-two per cent of the trials had a high complexity, 38% had a medium complexity, and 30% had a low complexity. Haematological trials had a higher mean score and a higher percentage of high-complexity studies. Phase I trials had a higher mean score and a higher percentage of high-complexity studies than phase 2 and phase 3 trials. No phase I studies fell into the low-complexity category.

## 4. Discussion

To the authors' knowledge, this is the first specific tool for assessing the complexity of pharmacy involvement in a sponsored oncology or haematology clinical trial. Pharm-CAT was found to be easy and user-friendly. The inter-pharmacist reproducibility of the assessment was confirmed in 77% of cases, and in 13 of the 14 cases where there was a difference in the final score, the two assessments placed the trial in the same complexity category, as the two scores did not straddle the subsequently established cut-offs, and Cohen's weighted kappa calculation confirmed agreement among the pharmacists.

The average score of the phase I trials was higher than those of the phase II and III trials, which was certainly due specifically to item 1 in Table 1 for which two extra points were awarded, but also to a general sensitivity of the tool. This emphasises the impact on the work of pharmacists when exceptions such as special drug preparation methods with dedicated devices or those not-yet compliant with the use of closed systems are encountered. We do not believe that there is a different sensitivity for Pharm-CAT in the evaluation of haematological and oncological studies; in fact, the differences found in the results were justified by a higher presence of haematological phase I studies (5 out of 21, or about 24%) than oncological studies (1 out of 39, or about 3%). The phase II trials recorded the lowest average score due to the standard trial design, which, in most cases, does not involve the administration of numerous experimental drugs or a blind, as there is no comparison arm.

Now that the complexity categories have been established with an initial sample of 60 clinical trials, it is necessary to continue evaluating the trials to verify the sensitivity of the cut-offs established. In particular, a second sample of trials is needed to confirm a homogeneous distribution among the categories and avoid a disproportionate number of trials falling unjustifiably into the same category. Our future goal is to analyse a larger set of assessments from different cancer research centres to verify the statistical concordance of the cut-offs established by this initial work and to make this tool widely available.

Differences in the regulatory requirements among countries may affect the general-isability, or there may be latent factors not captured by Pharm-CAT. Some activities were excluded because they were routinely performed (e.g., feasibility assessments, a site initiation visit, monitoring visits, and training activities). Users can modify Pharm-CAT to include items that are unique to their pharmacy service and exclude items that are not relevant or too specific. It is necessary to adapt the tool to the specific needs and resources of individual centres in order to measure the pharmacy workload in these settings.

## 5. Conclusions

Pharm-CAT proved to be user-friendly. The determination of complexity category cut-offs based on oncology- and haematology-sponsored trials led to the development of a simple and sensitive evaluation method in this field. Pharm-CAT aims to assess the workload as a prospective consideration when evaluating new clinical trials for activation and contracting, and to justify and negotiate trial pharmacy grants. In addition, the information obtained from the use of Pharm-CAT in a pharmacy service over time and at regular intervals will make it possible to monitor the evolution of the workload and the level of complexity of the work performed. Prospective multicentre validation of Pharm-CAT is needed to verify and confirm its applicability.

**Author Contributions:** Data curation, L.G., A.C., C.Z., C.A., P.S., G.B. and S.C.; investigation, L.G.; methodology, L.G., G.B. and S.C.; project administration, S.C.; supervision, C.M.; writing—original draft, L.G.; writing—review and editing, G.T., C.D. and S.C. All authors have read and agreed to the published version of the manuscript.

**Funding:** This research received no external funding.

**Institutional Review Board Statement:** Not applicable.

**Informed Consent Statement:** Not applicable.

**Data Availability Statement:** The data that support the findings of this study are not openly available due to reasons of sensitivity, and are available from the corresponding author upon reasonable request.

**Acknowledgments:** This work was partly supported thanks to the contribution of Ricerca Corrente by the Italian Ministry of Health within the research line "appropriateness, outcomes, drug value and organisational models for the continuity of diagnostic-therapeutic pathways in oncology".

**Conflicts of Interest:** The authors declare no conflicts of interest.

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
