# Peer review of "Pharmacy Workload in Clinical Trial Management: A Preliminary Complexity Assessment Tool for Sponsored Oncology and Haematology Trials"

_curroncol, doi:10.3390/curroncol31050218_

Round 1

Reviewer 1 Report

Comments and Suggestions for Authors

The authors presented an interesting study that uses a tool named Pharm-CAT, that categorizes it into ordinal complexity categories, to assess the complexity of clinical trials for consistent pharmacy grant applications. Some methodological aspects, in my opinion, need to be clarified and improved:

1 - there is no lack of work instructions or procedures regarding operations in these domains (and in the 2 centers) and therefore, it is not unknown that these operations are all carried out in the same way;

2 - there is no statistical support regarding the 3 domains studied; the strength of the classification variables and the answers given by professionals are not measured. For this reason, the results should be reviewed and discussion improved.

The table presented must be formatted (missing the bottom line) and captioned in lowercase letters.

Author Response

1 - there is no lack of work instructions or procedures regarding operations in these domains (and in the 2 centers) and therefore, it is not unknown that these operations are all carried out in the same way;

On page 3, method, lines 136-147 we have made the application of the individual scores more explicit, to make it clearer that these operations are all carried out in the same way.

2 - there is no statistical support regarding the 3 domains studied; the strength of the classification variables and the answers given by professionals are not measured. For this reason, the results should be reviewed and discussion improved.

We agree with this comment. Therefore, we have incorporated into the method (lines 157-158) and results (lines 176-178), the analysis of agreement between pharmacist raters through the application of Cohen's linear weighted kappa.

The table presented must be formatted (missing the bottom line) and captioned in lowercase letters.

We have corrected what was indicated.

Reviewer 2 Report

Comments and Suggestions for Authors

Thank you for allowing me to review this article of significant importance which focuses on Pharmacy workload in clinical trial management: a complexity 2 assessment tool for sponsored oncology and hematology trials.

The title is clear and coherent with the aim and content of the article.

Abstract

Well-structured summary with the essential information to give readers an overview of the article. 

Keywords

The chosen keywords are pertinent to the study, but they do not match the indexed or Mesh terms. This choice could make it challenging to display your article when searched in the database utilizing indexed terms, decreasing its recognition and dissemination.

Introduction

The introduction makes an explicit connection between the practical need of the field and the aim of the study. It could be reinforced with additional statistics or references that emphasize the inadequacy of current approaches.

When you use the acronym GCP you should describe what it means - line 37

Materials and Methods

Lines 114–116 state that the individual scores were agreed upon by four experienced pharmacists. It would be beneficial to understand on what basis there were four experts and no more.

Results

The results are presented in a summary and concise manner, demonstrating the applicability of the tool. It is also unclear how long the scale was applied and why the two centers where it was applied were chosen, given the difference in trials found between them.

Discussion

The document presents the points to be improved in a clear manner, as well as future aspects to be studied in order to apply the tool in general in the future. However, it could be supported by scientific evidence to justify these options, taking other tools used in the field as an example. 

Conclusion

Line 204 - 205: “This section is not mandatory but can be added to the manuscript if the discussion is unusually long or complex.” What is the intended meaning of this sentence in the conclusion? 

References

The references used are relevant to the type of study presented, but only 3 are from the last 5 years. Authors should try to support their work with recent and in-depth references, 11 references is a limited number. Other sources can help support the work presented. 

I want to congratulate the authors on this paper.

Author Response

Abstract: Well-structured summary with the essential information to give readers an overview of the article. 

Thank you

Keywords: The chosen keywords are pertinent to the study, but they do not match the indexed or Mesh terms. This choice could make it challenging to display your article when searched in the database utilizing indexed terms, decreasing its recognition and dissemination.

Thank you for your comment and we confirm that we have changed the keywords.

Introduction: The introduction makes an explicit connection between the practical need of the field and the aim of the study. It could be reinforced with additional statistics or references that emphasize the inadequacy of current approaches.

On page 2-3, lines 97-104, we have added additional comments to the main works already available.

When you use the acronym GCP you should describe what it means - line 37

We have made the acronym explicit

Materials and Methods: Lines 114–116 state that the individual scores were agreed upon by four experienced pharmacists. It would be beneficial to understand on what basis there were four experts and no more.

We have made this more explicit in lines 123-125. We felt that 4 pharmacists was the minimum number needed to construct a tool that took all aspects into account (2 pharmacists per participating centre). As this is a volunteer project, it is difficult to dedicate more people to this type of work without compromising the quality of the day-to-day work.

Results: The results are presented in a summary and concise manner, demonstrating the applicability of the tool. It is also unclear how long the scale was applied and why the two centers where it was applied were chosen, given the difference in trials found between them.

In lines 143-145 we have made the period of application explicit. The decision to develop and apply this tool in these two centres is based on their extensive experience in the field of oncological clinical research and the potential to take into account the aspects related to this activity.

Discussion: The document presents the points to be improved in a clear manner, as well as future aspects to be studied in order to apply the tool in general in the future. However, it could be supported by scientific evidence to justify these options, taking other tools used in the field as an example. 

We have added additional commentary to existing work (lines 97-104) to clarify our intentions and future goals (lines 209-211).

Conclusion: Line 204 - 205: “This section is not mandatory but can be added to the manuscript if the discussion is unusually long or complex.” What is the intended meaning of this sentence in the conclusion? 

Sorry, it's a typo in the initial template. Thank you for pointing it out, it has been removed.

References: The references used are relevant to the type of study presented, but only 3 are from the last 5 years. Authors should try to support their work with recent and in-depth references, 11 references is a limited number. Other sources can help support the work presented. 

We agree with the comment, but the subject is not well covered in the literature. We have tried to increase the number of references, but without improving the quality of this section. To justify our good work, one of the most recently reported papers (Song K. et al; October 2023) has comparable citations.

Thank you for the revision.

Reviewer 3 Report

Comments and Suggestions for Authors

The authors present a new tool for determining the complexities encountered by pharmacy departments when supporting clinical trials research. The manuscript is well written and the summary of prior research on the subject is comprehensive. The following comments are offered in order to strengthen the manuscript.

Title: considering adding "preliminary" before complexity because you mention that ", it is necessary to continue evaluating the trials to verify the sensitivity of the cut-offs established" at Line 184

Abstract:: You have no measurement or operational definition of the term "user friendly." How was this determined? It is not mentioned in your results. Either remove the sentence in the abstract, discussion and conclusion or provide a definition with qualitative and/or quantitative measurement.

Keywords: consider adding "hematology" and "workload" and changing "scoring tool" to "scoring method." Alphabetize the terms and use MeSH terms will increase searchability.

Methods: How was the data collected and tabulated? Why did you segment the results into thirds (0-33%, 34-66%, 67-100%) and how were they derived? Why didn't you include order set preparation (computer or manual) or patient type (inpatient or outpatient) or type of sponsor (investigator or industry-sponsored trail) as complexity factors? You hint at interrater reliability however consider using either Cohen's kappa or Cronbach's alpha to provide the reader with a better appreciation.

The numerical range of all possible scores needs to be calculated and reported.

Discussion: you mention several valid limitations with the tool. Consider including those potential factors that I mentioned above. Presumably, this scoring tool will allow for a better assessment of research costs. You could speculate on its ability to capture all research costs related to pharmacy delegation.

References need to be in MDPI style.

Thank you for the opportunity to review your very practical project.

Comments on the Quality of English Language

Minor grammar errors only that the copywriter will find.

Author Response

Title: considering adding "preliminary" before complexity because you mention that ", it is necessary to continue evaluating the trials to verify the sensitivity of the cut-offs established" at Line 184.

We agree with the observation and we have corrected the title.

Abstract:: You have no measurement or operational definition of the term "user friendly." How was this determined? It is not mentioned in your results. Either remove the sentence in the abstract, discussion and conclusion or provide a definition with qualitative and/or quantitative measurement.

On line 167, in the results section, we report on the ease of use of the tool (the clinical trial-trained pharmacist takes only 3-5 minutes to complete the table and assign the score). If you think it is misleading, we can delete the word ‘user friendly’.

Keywords: consider adding "hematology" and "workload" and changing "scoring tool" to "scoring method." Alphabetize the terms and use MeSH terms will increase searchability.

Thank you for your comment and we confirm that we have changed the keywords.

Methods: How was the data collected and tabulated?

We have included these details in lines 143-147.

Why did you segment the results into thirds (0-33%, 34-66%, 67-100%) and how were they derived?

As reported in lines 148-152, the construction of the tool did not automatically generate the complexity categories. However, having scored the first 60 trials, we decided to use the linear data from these 60 scores to construct the categories. The easiest way to do this was to sequence the 60 scores (run in duplicate for a total of 120 scores) and divide this sequence into tertiles. As you suggested, we used a statistical tool such as weighted Cohen's K to check the concordance of the scores and the assignment of the categories.

Why didn't you include order set preparation (computer or manual) or patient type (inpatient or outpatient) or type of sponsor (investigator or industry-sponsored trail) as complexity factors?

We have included these details in lines 143-147. The type of sponsor could not have been a factor in complexity, as the tool is constructed to evaluate only sponsored trials (as reported in the title and method). The type of patient has an impact as a general level of workload for a research centre, but has no specific impact on the work of the pharmacy and the management of the experimental drug.

You hint at interrater reliability however consider using either Cohen's kappa or Cronbach's alpha to provide the reader with a better appreciation.

We have included Cohen's linear weighted kappa as a measure of agreement between pharmacists.  

The numerical range of all possible scores needs to be calculated and reported.

We have included these details in lines 136-142.

Discussion: you mention several valid limitations with the tool. Consider including those potential factors that I mentioned above. Presumably, this scoring tool will allow for a better assessment of research costs. You could speculate on its ability to capture all research costs related to pharmacy delegation.

Thank you for your comments. We believe that in order to verify the economic impact of the tool, it is important to apply it in a concrete way, rather than through a simulation, in order to best highlight its potential. All this will be the subject of further work.

References need to be in MDPI style.

Thank you for your comment and we confirm that we have changed the references style.

Thank you for the revision.

Round 2

Reviewer 3 Report

Comments and Suggestions for Authors

The authors have addressed all my concerns and suggestions. Congratulations on the creation of a very practical tool for potential use in estimating the complexity of sponsored research trials.

Comments on the Quality of English Language

Minor issues that will be resolved in copywriting.